# Improving Risk Assessment for Metastatic Disease in Endometrioid Endometrial Cancer Patients Using Molecular and Clinical Features: An NRG Oncology/Gynecologic Oncology Group Study

**DOI:** 10.3390/cancers14174070

**Published:** 2022-08-23

**Authors:** Yovanni Casablanca, Guisong Wang, Heather A. Lankes, Chunqiao Tian, Nicholas W. Bateman, Caela R. Miller, Nicole P. Chappell, Laura J. Havrilesky, Amy Hooks Wallace, Nilsa C. Ramirez, David S. Miller, Julie Oliver, Dave Mitchell, Tracy Litzi, Brian E. Blanton, William J. Lowery, John I. Risinger, Chad A. Hamilton, Neil T. Phippen, Thomas P. Conrads, David Mutch, Katherine Moxley, Roger B. Lee, Floor Backes, Michael J. Birrer, Kathleen M. Darcy, George Larry Maxwell

**Affiliations:** 1Gynecologic Cancer Center of Excellence, Department of Gynecologic Surgery and Obstetrics, Uniformed Services University of the Health Sciences, Walter Reed National Military Medical Center, Bethesda, MD 20889, USA; 2The Henry M Jackson Foundation for the Advancement of Military Medicine, Inc., Bethesda, MD 20817, USA; 3Gynecologic Oncology Group Statistical and Data Management Center, Roswell Park Comprehensive Cancer Center, Buffalo, NY 14263, USA; 4Division of Gynecologic Oncology, Duke University, Durham, NC 27710, USA; 5Gynecologic Oncology Group Tissue Bank, Nationwide Children’s Hospital, Columbus, OH 43205, USA; 6Division of Gynecologic Oncology, University of Texas Southwestern Medical Center, Dallas, TX 75390, USA; 7Department of Obstetrics, Gynecology and Reproductive Biology, Michigan State University, 333 Bostwick Ave., NE, Grand Rapids, MI 49503, USA; 8Women’s Health Integrated Research Center, Women’s Service Line, Inova Health System, Falls Church, VA 22042, USA; 9Division of Gynecologic Oncology, Washington University, St. Louis, MO 63110, USA; 10Department of OB/GYN, Section of Gyn Oncology, University of Oklahoma University Health Sciences Center, Oklahoma City, OK 73104, USA; 11Department of GYN/ONC, Tacoma General Hospital, Tacoma, WA 98405, USA; 12Division of Gynecologic Oncology, Ohio State University Comprehensive Cancer Center, Columbus, OH 43210, USA; 13P. Rockefeller Cancer Institute, Women’s Gynecologic Cancer Clinic, Little Rock, AR 72205, USA

**Keywords:** endometrial cancer, metastasis, prediction, molecular classifier, risk assessment

## Abstract

**Simple Summary:**

Judging the chance (risk) of cancer spread and bad outcome in endometrial cancer continues to be a challenge. Molecular and clinical factors offer the hope of improving the accuracy of judging the danger of cancer spread (metastasis) and a bad outcome (prognosis) to help guide patient care. The aim of this research was to develop a risk score for cancer spread and bad outcome for the most common type of endometrial cancer, endometrioid endometrial cancer, using molecular features and clinical factors in endometrial cancers removed during surgery. The molecular score, referred to as MS7, was more accurate at judging the chance of nodal and distant metastasis than clinical factors like grade 3 disease and myometrial invasion. MS7 score was also better than aggressive molecular subtypes or endometrial cancer-associated genes identified by other research groups. The combination of MS7 score and myometrial invasion was the best at accurately judging the chance of nodal and distant metastasis in the most common type of endometrial cancer. The MS7 score was also shown to accurately indicate bad outcome including cancer progression and death. This research hopes to help guide patient care stopping overtreatment in lower-risk and undertreatment in higher-risk endometrial cancer patients.

**Abstract:**

**Objectives:** A risk assessment model for metastasis in endometrioid endometrial cancer (EEC) was developed using molecular and clinical features, and prognostic association was examined. **Methods:** Patients had stage I, IIIC, or IV EEC with tumor-derived RNA-sequencing or microarray-based data. Metastasis-associated transcripts and platform-centric diagnostic algorithms were selected and evaluated using regression modeling and receiver operating characteristic curves. **Results:** Seven metastasis-associated transcripts were selected from analysis in the training cohorts using 10-fold cross validation and incorporated into an MS7 classifier using platform-specific coefficients. The predictive accuracy of the MS7 classifier in Training-1 was superior to that of other clinical and molecular features, with an area under the curve (95% confidence interval) of 0.89 (0.80–0.98) for MS7 compared with 0.69 (0.59–0.80) and 0.71 (0.58–0.83) for the top evaluated clinical and molecular features, respectively. The performance of MS7 was independently validated in 245 patients using RNA sequencing and in 81 patients using microarray-based data. MS7 + MI (myometrial invasion) was preferrable to individual features and exhibited 100% sensitivity and negative predictive value. The MS7 classifier was associated with lower progression-free and overall survival (*p* ≤ 0.003). **Conclusion:** A risk assessment classifier for metastasis and prognosis in EEC patients with primary tumor derived MS7 + MI is available for further development and optimization as a companion clinical support tool.

## 1. Introduction

The incidence and mortality of endometrial cancer continue to increase, with persistent disparities in outcomes [1]. Surgical and pathologic staging has been advocated as the gold standard for risk assessment and treatment planning in endometrial cancer. However, most patients with presumed uterine-confined disease do not have lymph node metastasis [1]. Five-year survival for endometrial cancer patients is 95% when disease is diagnosed at a local stage and drops to 68% with regional spread or to 17% with distant metastasis [2]. Elimination of unnecessary lymph node assessment could reduce intra- and postoperative patient morbidity. Previously studied clinicopathologic factors, imaging, and molecular biomarkers have failed to make binary predictions regarding lymphatic metastasis [3,4,5,6,7,8,9,10,11,12,13,14,15,16,17]. Efforts to advance risk stratification in endometrial cancer, most notably in the lower risk endometrioid subtype, are needed to guide patient care and reduce both overtreatment and undertreatment in this disease setting.

Molecular and clinical factors offer the promise of enhanced risk stratification of metastasis and prognosis with sufficient precision to help guide endometrial cancer patient care. The Cancer Genome Atlas (TCGA) consortium identified endometrial cancer subtypes based on unique molecular profiles [18,19], and some were associated with aggressive phenotypes and poor clinical outcomes. Patients with endometrioid endometrial cancers were found to have a higher prevalence of POLE mutations, microsatellite instability, and low somatic copy number alterations. Previous work by our group suggested that transcript expression in primary endometrioid endometrial cancers is unique between patients with and those without metastasis [20]. These studies underscore the potential for a molecular classifier to stratify risk of metastasis, recurrence, progression, and prognosis with performance characteristics rigorous enough for clinical utilization.

We developed a promising molecular classifier of risk and prognosis for the most common type of uterine adenocarcinoma, endometrioid endometrial cancer (EEC), using molecular and clinical assessments in hysterectomy-excised primary tumors. The transcript-based classifier of metastasis, referred to as MS7, exhibited the best prediction of nodal and distant metastasis compared with grade 3 disease (G3), myometrial invasion categorized at 50% or greater (MI), three aggressive molecular subtypes defined by the Uterine Corpus Endometrial Carcinoma (UCEC) Research Network of TCGA [19], and both overexpression and mutation of a panel of endometrial cancer-associated genes in the TCGA Training-1 cohort. A risk assessment model for metastasis based on the MS7 classifier and MI predicted risk of nodal and distant metastasis in EEC patients with stage I, IIIC, or IV disease with a sensitivity and negative predictive value (NPV) of 100% in two validation cohorts. This investigation achieved its primary objective of developing and validating a promising risk assessment tool for EEC patients based on molecular and clinical features in the primary uterine tumor. The exploratory component of this study provided the distribution of a primary tumor-derived MS7 score in stage I–IV EEC patients, demonstrating its potential prognostic value. The ultimate goal of the risk assessment model is to augment pathologic assessments, including intraoperative frozen section and sentinel node dissection, with molecular features to guide clinical management, avoiding either overtreatment in lower-risk cases or undertreatment in higher-risk endometrial cancer patients.

## 2. Materials and Methods

### 2.1. Study Design and Characteristics

This study was performed under exempt retrospective protocols for molecular profiling of endometrial cancer (GOG-8024/15-1806 and 14-1679). The primary objective of this investigation was to develop/train and test/validate a risk assessment model for nodal and distant metastasis using RNA-sequencing (RNAseq) or hybridization-based (microarray) transcriptomic data and clinicopathologic characteristics. We utilized a case–control design to address this objective. EEC patients with stage IIIC/IV disease were classified as cases, and EEC patients with stage I disease were designated as controls. Cases and controls with RNAseq data were aggregated into the Training-1 and Validation-1 cohorts, whereas those with microarray data were assigned to the Training-2 and Validation-2 cohorts. This study also incorporated an exploratory component to describe the stage distribution and potential prognostic value of the transcript-based model in EEC patients with stage I–IV disease with existing RNAseq data, including patients in the Training-1 and Validation-1 cohorts.

RNAseq and clinical data were downloaded for 389 EEC patients from the UCEC TCGA Research Network [18]. The exploratory component of this study combined the 75 patients in the Training-1 cohort and 230 patients in the Validation-1 cohort with 22 additional patients with stage IB disease (source: GOG) and 62 patients with stage II, III not specified, IIIA, or IIIB disease from TCGA. The GOG, currently known as NRG Oncology, provided clinical data and frozen primary tumors for 64 EEC patients in the Training-2 cohort, and we generated hybridization-based Affymetrix Plus 2.0 microarray data for these cases and controls. We also contributed existing RNAseq and limited pathology data for the Validation-1 cohort (N = 15) [21], as well as existing hybridization-based microarray data and limited pathology data for the Validation-2 cohort (N = 81) [22,23].

The stage I controls in the Training-1 and Training-2 cohorts were required to have undergone strict pelvic and para-aortic lymph node sampling and to be alive at last contact with no evidence of disease after three or more years of follow up. The criteria for lymph node sampling were adopted from the GOG-210 protocol and required histologic evaluation of at least four left and four right pelvic lymph nodes, as well as one left and one right para-aortic lymph node. In contrast, the eligibility criteria for the stage I cases in the two validation cohorts were less stringent for practical reasons, accepting controls that did not satisfy the strict criteria required for the training cohorts.

### 2.2. Transcript Expression Data

Upper-quantile normalized level 3 RNAseq V2 RSEM expression data generated by the UCEC Research Network were acquired from the TCGA data portal (https://tcga-data.nci.nih.gov/, on 19 February 2014, and now maintained at https://gdc.cancer.gov/). The normalized RNAseq data for 20,531 genes were log_2_-transformed (log_2_(RSEM + 1)), and those with detectable RSEM levels > 0 in at least 2/3 of samples used for the primary objective were selected for further evaluation. Upper-quantile normalized RNAseq data generated by our group, the GYN-COE, were repurposed for this investigation [21] using methods included in the first section of the Appendix A for 15 EEC patients incorporated into the Validation-1 cohort. Robust multiarray average (RMA) normalized Affymetrix Plus 2.0 microarray data for this investigation were generated and processed by the GYN-COE [22,23] as described in in the second section of the Appendix A. Briefly, the RMA algorithm implemented in the ‘*simpleaffy*’ Bioconductor package was applied to raw CEL files, and only those probe sets detected with expression levels > 0 in at least 2/3 samples were retained for further analysis. Top RNAseq-based candidates were mapped to HG-U133 Plus 2.0 annotations downloaded from https://www.ncbi.nlm.nih.gov/geo/query/acc.cgi?acc=GPL570, on 19 February 2014.

### 2.3. Transcript Selection and Classifier Development

The third and fourth sections of the Appendix A provide detailed methods for screening, selection, and evaluation of RNAseq- and microarray-based transcripts and multitranscript classifiers of nodal and distant metastasis. Briefly, logistic regression modeling with randomized subsampling in R was used to rank and prioritize the selection of robust transcripts with a strong and consistent relationship with nodal and distant metastasis in the training cohorts. Platform-centric multivariate logistic modeling for metastasis was exhaustively performed using WEKA to evaluate each of the unique combinations of the candidate transcripts from the previous step. Fixed platform-centric diagnostic algorithms were generated in SAS for the multitranscript classifier (Figure 1) using the following formula with coefficients derived from platform-centric multivariate logistic regression modeling.
$$MS7\;Risk\;Score\;for\;Metastasis=\sum_{i=1}^{n}\left(transcript\;expression\;i\;x\;coefficient\;i\right)$$

Logistic regression modeling and ROC curve analysis were implements using SAS to evaluate the relationship and the accuracy of prediction of nodal and distant metastasis in the various cohorts.

### 2.4. Relationships with Cancer Biomarkers and Functional Pathway Analysis

The fifth section of the Appendix A provides details about the methods used to examine the relationship between categorical variables using Fisher’s exact test and logistic regression modeling, the correlation between continuous variables using Spearman’s rank test, and the identification and functional pathway analysis of differentially expressed transcripts in Training-1 patients with the highest score (quartile 4) compared with the those with the lowest score (quartile 1).

### 2.5. Exploring the Prognostic Relationship between MS7 and Clinical Outcome

Survival analyses for time-to-event data were performed in SAS using univariate Cox proportional hazard regression modeling with a Wald test and the Kaplan–Meier method with a log-rank test in 389 EEC patients with stage I–V disease from TCGA and in the subset of TCGA cases categorized by stage. Subset analyses in the Training-1, Validation-1, or Training-2 cohort were also included. Cox modeling with adjustment for stage or adjustment for age, stage, G3 disease, and ≥50% MI was performed in the full set of 389 EEC patients from TCGA. Censoring of events was performed 60 months after diagnosis. Progression-free survival (PFS) was calculated as the time in months from diagnosis to disease progression or death or to the date of the last contact for women who were alive with no evidence of disease progression (censored). Overall survival (OS) was calculated as the time from diagnosis to death or to the date of the last contact for those who were still alive (censored).

## 3. Results

The primary objective of this study was to develop a companion diagnostic model for nodal and distant metastasis for EEC patients using molecular and clinical features available at the time of a hysterectomy. Table 1 summarizes the clinical characteristics and source for the EEC patients in this investigation grouped by type of transcriptomic data and platform-centric cohort. The Training-1 cohort included 29 stage I and 46 stage IIIC/IV cases from the UCEC TCGA Research Network with existing RNAseq and clinical data [18]. The Validation-1 cohort integrated 225 stage I and 5 stage IIIC/IV cases from the UCEC TCGA Research Network [19] with 8 stage I and 7 stage IIIC/IV cases from the GYN-COE with existing RNAseq and clinical data [21]. The Training-2 cohort was composed of 31 stage I and 33 stage IIIC/IV cases with clinical data from the GOG-210 protocol and microarray data generated by the GYN-COE for this investigation. The Valdiation-2 cohort was made up of 69 stage I and 12 stage IIIC/IV EEC cases with microarray data previously generated by the GYN-COE [22,23] and repurposed for this study. The 15 patients added to the Validation-1 cohort and the 81 patients in the Validation-2 cohort from the GYN-COE had limited clinical annotation, often restricted to site of disease, histology, stage, grade, and myometrial invasion.

### 3.1. Selecting Transcripts Associated with Metastasis

A total of 17,265 RNAseq-derived transcripts were individually screened to identify the 1630 transcripts associated with nodal and distant metastasis in the 75 Training-1 patients, with *p* < 0.05 (Appendix A). This list was then trimmed using 100 randomly selected subsamples of Training-1 to select the 311 RNAseq-derived transcripts with robust performance in at least 80 of the subsamples (Appendix A), including 268 transcripts that mapped to an Affymetrix probe set and were prioritized for further evaluation. Of the 268 transcripts, 33 were associated with metastasis, with *p* < 0.05, in the 64 Training-2 cases (Appendix A), and 23 of these transcripts exhibited a consistent positive or negative univariate relationship with nodal and distant metastasis in both the Training-1 and Training 2 cohorts (Appendix A). Higher levels of 11 of these transcripts indicated a higher risk of nodal or distant disease and a positive coefficient in the logistic regression model. Higher levels of 12 transcripts indicated a lower risk of nodal or distant metastasis and a negative coefficient in the logistic regression model.

### 3.2. Developing a Transcript-Based Classifier of Metastasis

Using 10-fold cross validation in the training cohorts, an exhaustive combinatorial analysis of the top 23 transcripts was performed to identify the subset of biomarkers that provided the best prediction parameters. Briefly, the four-step selection process yielded 39,788 multitranscript classifiers during step 1, 27 during step 2, 7 during step 3, and the top multitranscript classifier during step 4. The constituents in the top multitranscript classifier are displayed in Figure 1. The platform-centric algorithms for the multitranscript classifier referred to as MS7 are also presented in Figure 1.

### 3.3. Evaluating a Transcript-Based Classifier of Metastasis with Other Features

Table 2 illustrates the accuracy of MS7 ± G3 ± MI in terms of predicting metastasis in the Training-1, Validation-1, Training-2, and Validation-2 cohorts. Age at diagnosis and the molecular features listed below were also evaluated for their ability to predict nodal and distant metastasis in the Training-1 cohort (Appendix A). This included assessments of the three aggressive molecular subtypes defined by the UCEC TCGA Research Network (copy number variant high (CNV) subtype, somatic copy number alterations (SCNA) cluster 4 subtype, and the transcript-based mitotic molecular subtype) [19]; RNAseq-based transcript expression data for *ESR1*, *ARID1A*, *CTNNB1*, *KRAS*, *MKI67*, and *PIK3CA*; and mutation status in TP53 or PTEN.

MS7 was the best predictor of nodal and distant metastasis in the Training-1 cohort, with an AUC (95% CI) of 0.89 (0.80–0.98) as a continuous variable or 0.87 (0.97–0.96) as a categorical variable, followed by 0.69 (0.59–0.80) for MI, 0.66 (0.55–0.77) for G3 disease, and 0.71 (0.58–0.83) for *ESR1* transcript expression. Age at diagnosis and the other molecular features were then dropped from further consideration (Appendix A). Next, we compared the accuracy of diagnostic models that combined MS7 score ± G3 ± MI to predict nodal and distant metastasis in the Training-1 cohort. The AUC (95% CI) was 0.89 (0.80–0.98) for both the MS7 + G3 combination and the model with MS7 alone and 0.92 (0.85–0.99) for both the MS7 + MI combination and the MS7 + MI + G3 combination. Transcript expression of *ESR1* exhibited a higher univariate AUC than that of G3 or MI but did not improve the predictive accuracy when combined with MS7 (AUC 0.89, 95% CI: 0.81–0.97) or when combined with MS7 and MI (AUC 0.92, 95% CI: 0.86–0.99) (Appendix A).

The MS7 ± G3 ± MI classifiers were then evaluated for their univariate and multivariate accuracy in predicting nodal and distant metastasis in the Validation-1, Training-2, and Validation-2 cohorts (Table 2 and Appendix A). MS7 was a better univariate predictor of nodal and distant metastasis than G3 disease or MI ≥ 50% in the Validation-1 and Training-2 cohorts and was second to MI ≥ 50% as a metastasis predictor in the Validation-2 cohort. The MS7 + MI + G3 combination exhibited a similar predictive accuracy (AUC, 95% CI) to that of the MS7 + MI combination in the Validation-1 cohort (0.83, 0.72–0.94 vs. 0.81, 0.71–0.92), the Training-2 cohort (0.92, 0.86–0.99 vs. 0.91, 0.84–0.98), and the Validation-2 cohort (0.87, 0.78–0.97 vs. 0.86, 0.77–0.96).

### 3.4. Developing and Evaluating Diagnostic Models of Metastasis

ROC analysis was then applied to compare the performance of the companion diagnostic models using the MS7 risk score categorized using optimal platform-centric cut points alone or combined with either G3 or MI (Table 3). The optimal cut point for the MS7 risk score was −2.00356 using the RNAseq-based algorithm and −4.25324 using the microarray-based algorithm. This enabled us to assess performance in Validation-1 (N = 245), Validation-2 (N = 81), and the merged Validation-1+-2 cohorts (N = 326) based on data summarized in Appendix A. MS7 had a sensitivity of 0.79 (95% CI = 0.58–0.93) and a negative predictive value (NPV) of 0.94 (95% CI = 0.90–0.98) for a negative test indicated by a low platform-centric MS7 risk score. The addition of G3 or MI to the MS7 risk score enhanced the accuracy and potential utility to distinguish EEC patients with nodal or distant metastasis vs. stage I disease. The combination of MS7 and G3 disease had a sensitivity of 0.88 (95% CI = 0.68–0.97) and an NPV of 0.95 (95% CI = 0.90–1.00), correctly classifying 137/140 validation cases, with a negative test indicated by a low MS7 score and grade 1 or 2 disease. In contrast, the combination of MS7 and MI had 100% sensitivity and NPV in this cohort, correctly classifying 145/145 validation cases, with a negative test result indicated by a low MS7 risk score and <50% MI.

### 3.5. Evaluating the Biologic Plausibility of the MS7 Classifier of Metastasis

Analyses were then performed to investigate the relationship between the MS7 classifier, the three aggressive molecular subtypes defined by the UCEC TCGA Research Network, cancer biomarkers, and functional pathways. The UCEC TCGA Research Network defined three aggressive molecular subtypes: CNV high, SCNA cluster 4, and transcript-based mitotic subtype [19]. High vs. low MS7 in EEC patients with stage I, IIIC, or IV disease evaluated for molecular subtypes by TCGA were more likely to present with the SCNA cluster 4 subtype (OR 2.839, 95% CI: 1.257–6.410, *p* = 0.012) or with the mitotic subtype (OR 9.606, 95% CI: 4.491–20.545, *p* < 0.0001). However, the CNV high subtype was rare in this set of patients (13 of 155 patients, including 5 with low MS7 and 8 with high MS7).

Higher levels of MS7 were correlated with higher expression of *ARID1A*, *CTNNB1*, *KRAS*, *MKI67,* and *PIK3CA* and lower expression of *ESR1* in both the Training-1 (N = 75) and Validation-1 (N = 245) cohorts. These relationships were highly statistically significant, but correlation coefficients did not exceed 0.396 for the direct relationships noted above or −0.421 for the inverse relationship with *ESR1* (Appendix A). A total of 197 EEC patients from TCGA were evaluated for mutation status. In this subset, the proportion of patients with high vs. low MS7 risk score varied depending on *CTNNB1* mutation or *TP53* mutations status but not depending on mutations in ARID1A, KRAS, PIK3CA, or PTEN (Appendix A). Patients with high vs. low MS7 were 64% less likely to exhibit a mutation in *CTNNB1* (odds ratio (OR): 0.438, 95% confidence interval (CI): 0.231–0.832, *p* = 0.012) and 4.4 times more likely to have a mutation in *TP53* (OR: 4.415, 95% CI: 1.862–10.470, *p* = 0.007). The relationships observed between the categorized MS7 risk score and the aggressive molecular subtypes as defined by the Cancer Genome Atlas (TCGA) Research Network are shown in Appendix A.

Differential transcript expression analyses were performed in patients classified as exhibiting a high MS7 (upper quartile (Q4), n = 19) versus a low MS7 (lower quartile (Q1), n = 19) risk score for nodal or distant metastasis in the Training-1 cohort, and subsequent alterations in canonical molecular pathways were determined. A total of 1785 transcripts were significantly and differentially abundant between Q4 and Q1 patients (q-value ≤ 0.05); the subset with a log2 ratio of gene expression less than −2 or greater than +2 is presented in Appendix A. There were 25 enriched molecular pathways with significant differentially expressed genes between Q4 and Q1 patients, including pathways involved in DNA replication and repair, inflammatory and estrogen signaling, and regulation of protein translation and stability, as shown in Figure 2 and Appendix A, with percent pathway enrichment displayed in Appendix A.

### 3.6. Exploring the Potential Prognostic Value of MS7

Appendix A illustrates the stage distribution for the MS7 risk score in the 389 EEC patients evaluated by the UCEC TCGA Research Network. These patients had stage I–IV disease. EEC patients with stage I-IIB disease displayed a range of MS7 scores, whereas most of the patients with stage IIIC, or IV disease selectively exhibited a high MS7 score. Figure 3 displays PFS in EEC categorized into high or low MS7 risk score using the platform-centric cut points (−2.00356 for the RNAseq-based algorithm and −4.25324 for the microarray-based algorithm).

High vs. low MS7 indicated worse PFS (Figure 3A, log-rank test. *p* < 0.001) and an increased risk of disease progression (hazard ratio (HR) = 2.51, 95% CI = 1.59–3.95) in the 347 stage I–IV EEC patients from TCGA. This relationship between MS7 and PFS held up to an adjustment for stage alone (adjusted HR = 1.87, 95% CI = 1.13–3.07) or adjustments for age, stage, G3, and MI (adjusted HR = 2.12, 95% CI = 1.19–3.78). The relationship between high vs. low MS7 and PFS remained significant in the subset of EEC patients with stage I, IIIC, or IV disease in the training cohorts (Figure 3B, log-rank test, *p* = 0.003; Figure 3C, log-rank test, *p* < 0.001) and in stage II–IV EEC patients from TCGA (Figure 3F, log-rank test, *p* < 0.001). A trend suggesting worse PFS was observed in the subset of Validation-1 cases and in the stage I EEC cases from TCGA, as shown in Figure 3D,E, respectively.

Figure 4 displays OS in EEC categorized into high or low MS7 risk score. High vs. low MS7 indicated significantly worse OS in the 389 stage I–IV EEC patients from TCGA (Figure 4A, log-rank test, *p* = 0.002) and an increased risk of death (HR = 2.48, 95% CI = 1.35–4.56). The relationship between high vs. low MS7 risk score and worse OS was downgraded to a trend after adjusting for stage (adjusted HR: 1.63 (0.83–3.21)) or for age, stage, G3, and MI (adjusted HR: 1.44, 95% CI: 0.65–3.22). Worse OS persisted in the subset of EEC patients with stage I, IIIC, or IV disease in the two training cohorts (Figure 4B, log-rank test, *p* = 0.016; Figure 4C, log-rank test, *p* = 0.002) but was not observed in the 230 Validation-1 cases from TCGA (Figure 4D). Follow-up data were not available for the 15 patients from the GYN-COE in the Validation-1 cohort or for the 81 patients in the Validation-2 cohort. Appendix A shows the available adjuvant treatment, PFS, and OS data for these cohorts. Appendix A illustrate the independent dominance of the MS7 risk score using adjusted Cox models of risk of disease progression with both classic clinical factors and endometrial cancer biomarkers. 

## 4. Discussion

Several investigations have examined the relationship between specific molecular alterations and endometrial cancer behavior, but few have specifically focused on the association of molecular alterations with endometrial cancer metastasis [13,24,25,26,27,28,29]. Previous efforts assessed the utility of a biomarker panel to predict metastasis in a uterine curettage (e.g., ER/PR loss or TP53, bcl-2, Her-2, MIB-1, and PCNA) [25,27]. Histologic subtype and immunohistochemical expression of TP53 or bcl-2 strongly correlated with lymph node spread. However, in multivariate regression, only TP53 predicted metastasis. Engelsen et al. [29] reported an association between immunohistochemical overexpression of TP53 or p16 in preoperative curettage specimens and advanced-stage disease (*p* < 0.001). To date, the accuracy of these biomarkers in identifying patients with extrauterine disease is equivalent to that of traditional pathologic variables, specifically grade and depth of invasion [3,4,5,6,7,8,9,10,11,12,13,14,15,16,17]. More recent efforts to predict metastasis in endometrial cancer patients have incorporated clinical and/or molecular features [29,30,31,32,33,34,35,36,37,38,39,40,41,42,43].

In preliminary work by our group, we documented differences in transcript expression in primary tumors from patients with node metastasis compared with disease confined to the uterus [20]. We then hypothesized that a companion diagnostic model based on transcripts ± G3 ± MI may be more accurate than pathologic features in the primary tumor for identifying metastasis. We focused our investigation on EEC, as this is a patient population inconsistently referred for specialty care and at high risk for overtreatment. Stage I controls in the training cohort were required to undergo strict surgical stage and to be recurrence-free with at least 3 years of follow-up to avoid stage I recurrences that might reflect an unrecognized micro-metastasis. However, the stage I controls in the validation cohorts included patients with less stringent criteria for staging, recurrence status, and follow-up time. The RNAseq data from the TCGA were generated using intact frozen primary tumors [18]. In contrast, the RNAseq and the microarray data generated by the GYN-COE program were derived from enriched frozen primary tumors following laser microdissection or macroscopic scraping [21,22,23]. The utilization of a subset of cases and controls with transcriptomic data derived from samples enriched with tumor cellularity helped prioritize the selection of tumor-cell-associated transcripts.

In our study, we developed and validated candidate transcripts and a multitranscript classifier using both RNAseq and microarray data. The MS7 classier distinguished EEC patients with nodal and distant metastasis from stage I disease when evaluated as a continuous score or using platform-centric cut points. Improvements in diagnostic accuracy were observed by combining MI with the MS7 score. The MS7 + MI combination exhibited an AUC (95% CI) of 0.81 (0.71–0.92) in the Validation-1 cohort and 0.86 (0.76–0.96) in the Validation-2 cohort. Bendifallah et al. [44] recently illustrated the AUC (95% CI) for clinical predictors of lymph node metastasis: 0.69 (0.66–0.72) for PORTEC-1, 0.69 (0.67–0.71) for GOG-99, 0.68 (0.66–0.70) for SEPAL, 0.70 (0.68–0.72) for EMSO, and 0.80 (0.78–0.82) for EMSO modified using a three-tier risk classification ± LVSI. Given the need to minimize misclassification of patients with nodal and distant metastasis, we determined that the NPV (95% CI) for the combination of MS7 + G3 was 0.98 (0.94–0.996), with correct classification of 137 of 140 negative tests, as indicated by low MS7 and grade 1 or 2 disease. The NPV for the combination of MS7 + MI was 1.00, with correct classification of 145 of the 145 negative tests, as indicated by low MS7 and <50% MI. Trovik et al. demonstrated that loss of ER and PR had an NPV of 0.92, whereas pathologic TP53 expression had an NPV of 0.91 [25]. The 12-transcript-based prediction signature for nodal metastasis in EEC developed by Kang et al. using principal component analysis displayed an NPV of 100%, a specificity of 41%, and a PPV of 21% in a validation cohort including 108 patients [29]. The NPV, specificity, and PPV for our seven-transcript-based signature of metastasis with MI were 100%, 49%, and 26%, respectively, in two validation cohorts representing 326 patients. There was no overlap in the transcripts incorporated into these two signatures, and our signature differed from that developed by Kang et al. by incorporating MI with the transcripts.

A companion diagnostic tool based on G3 ± MI is clearly inferior to that based on MS7 + MI. Additional research is needed to evaluate the MS7 + MI model in expanded cohorts of independent cases to determine whether other clinical and/or molecular assessments beyond those evaluated in the Training-1 cohort may further enhance the predictive accuracy of MS7 + MI. Transitioning the MS7 assessment from a frozen to a formalin-fixed and paraffin-embedded tumor sample and from a high-throughput “omic” platform to a quantitative target-based assay, such as a quantitative RT-PCR assay, represent other important steps to advance the development and deployment of this companion diagnostic. Additional research is needed to evaluate the performance of the MS7 classifier as a companion diagnostic and prognostic tool in a diagnostic biopsy or in a metastatic tumor sample.

Identification of a more accurate test for prediction of metastasis has clinical implications for management of endometrial cancer. Although surgical staging by a gynecologic oncologist is recommended, only 24% of women with newly diagnosed endometrial cancer are referred to such subspecialists [45]. When compared to clinical care provided by gynecologists and other subspecialists, management by a gynecologic oncologist has been associated with a higher incidence of surgical staging and improved patient survival among patients found to have metastatic disease [45,46]. If general gynecologists had a more accurate test on which to base referrals, we speculate that a larger proportion of newly diagnosed endometrial cancer patients would be properly referred. In addition, information from an MS7-like test in un-staged cases referred to a specialist would contribute to a more personalized approach when deciding between surveillance, interval surgical staging, and empiric radiotherapy. Unfortunately, variations in practice patterns among gynecologists and subspecialty gynecologic oncologists highlight the need for biomarkers to incrementally enhance conventional pathologic features for risk assessment [47,48]. We were reassured by the improved sensitivity and the high NPV of the MS7 risk score, especially when combined with MI, compared with that indicated by G3 disease, which is currently used by many gynecologists to make clinical decisions regarding referral. Gynecologic oncologists may potentially opt for sentinel node biopsy instead of the more aggressive pelvic and para-aortic nodal dissection in EEC patients with a low MS7 score and <50% MI. Alternatively, patients with a high MS7 score and ≥50% MI may be candidates for extensive pelvic and para-aortic nodal excision, and some may require adjuvant treatment.

Previous analyses demonstrated the cost-effectiveness of a screening test for endometrial cancer metastasis. Testing remained cost effective (<$50,000/life year saved) unless the rate of referral to a gynecologic oncologist for full staging exceeded 90% [49]. Given the current low rates of full surgical staging by generalists and/or referral to a gynecologic oncologist, a diagnostic test to detect nodal metastasis for endometrial cancer has potential to be cost-effective, in addition to optimizing patient outcomes. Although tumor assays, such as Mammostrat, Oncotype DX, and MammaPrint, are clinically useful in prediction of breast cancer outcomes, no such pretreatment tests are currently available for endometrial cancer.

We acknowledge several limitations of our study. First, we recognize that differences in methods used to prepare the samples and analyze the cases existed between the TCGA (Training-1) and the GOG (Training-2) datasets, which limited the feasibility of using one for discovery and the other for validation. Second, the number of cases with metastatic disease in our investigations limited our ability to control for false discovery using q-values based on Benjamini and Hochberg or Storey. Modeling with resampling in 80% of Training-1 cases repeated 100 times and utilization of 10-fold cross validation of Training-1 and -2 cohorts for selection of candidate transcripts and classifiers helped us reduce the likelihood of making selections by chance. Validation of the MS7 risk score in independent cases provided statistical evidence of the association between MS7 and metastasis. Exclusion of GOG cases from the Validation-1 and -2 cohorts helped ensure independence between the training and validation cohorts. Third, the quality of tumors for analysis of transcript expression using samples from GOG may have presented a selection bias, given the fact that less than half of cases passed quality assurance analysis (RIN > 5). Fourth, although the number of advanced-stage cancers was low in the validation cohorts, we specifically expanded the numbers of stage I cancers to provide reassurances regarding negative predictive value, which was shown to be 100% in our analyses. Fifth, modification of the selection methods yields alternate predictors of metastasis, but none of these classifiers consistently outperformed MS7 in these cohorts (data not shown). However, we provided evidence indicating the potential clinical value of a promising molecular diagnostic model based on MS7 + MI. As described above, additional efforts are required to further strengthen the performance of this model and to transition from an “omic”-based MS7 assessment to a clinical test performed in archival formalin-fixed and paraffin-embedded samples.

The MS7 signature includes an antisense long noncoding RNA (lncRNA) candidate (BDNF-AS) and six protein-coding genes with diverse cellular functions. Brain-derived neurotrophic factor antisense RNA (BDNF-AS) is an endogenous lncRNA transcribed from the BDNF genomic loci that can directly regulate BDNF gene expression [50,51]. Elevation of BDNF has been associated with poor prognosis in neuroblastoma [52] and has been shown to increase tumor cell viability via BDNF-mediated activation of tropomyosin receptor kinase B (TrkB) signaling in gynecologic cancer cells [53]. In uterine cancers, BDNF and TrkB protein levels have been shown to be elevated in endometrial cancer versus normal tissues, with elevated TrkB correlating with increased lymph node metastasis and lymphovascular space involvement [54]. BDNF/TrkB signaling has directly been shown to promote epithelial-to-mesenchymal transition and to inhibit anoikis in endometrial carcinoma cells both in vitro and in vivo [54,55]. Apolipoprotein 4 (APOL4) is a member of the apolipoprotein L family that regulates cellular lipid homeostasis, as well as cellular immunity and programmed cell death signaling [56,57]. Our observed loss of BDNF-AS in metastatic patients is consistent with these previous findings. To that end, differential gene expression analyses (described below) revealed a non-significant yet suggestive trend of elevated BDNF gene expression in metastatic patients (BDNF, 1.24 Log_2_ FC Q4 vs. Q1, q-value = 0.57, data not shown). Myeloid/lymphoid or mixed-lineage leukemia translocated to 10 (MLLT10) is a putative transcription factor that commonly presents as a gene fusion product with clathrin assembly lymphoid myeloid (CALM) in T-cell leukemia [58]. The resulting fusion protein impacts hematopoietic cell differentiation via dysregulation of homeobox A gene expression [58]. PDZ and LIM domain 3 (PDLIM3) is a cytoskeletal protein that has been shown to regulate actin organization, and alternatively spliced variants of this gene have been observed in skeletal muscle from muscular dystrophy type 1 patients [59,60,61]. In cancer, PDLIM3 has been identified as a candidate in a five-gene signature that can predict activated hedgehog signaling in medulloblastoma patient tissues [62]. Arginine/serine-rich coiled-coil 1 (RSRC1) functions in canonical and alternative mRNA splicing activities [63] and has further been shown to regulate SUMOylation and the downstream transcriptional activity of estrogen receptor 2 [64]. Transforming growth factor beta regulator 1 (TBRG1) is a chromatin-associated protein with tumor-suppressor-like activities that mediates TP53 activation via diverse mechanisms, including interaction with the ARF tumor suppressor (alternative reading frame product of the CDKN2A locus) and TP53 antagonist MDM2 (mouse double minute 2 homolog), as well the histone acetyltransferase KAT5 (Tip60), promoting G1-phase cell cycle arrest and chromosomal instability in tumor cells [65,66]. Zinc finger protein 596 (ZNF596) is a putative zinc-finger binding transcription factor (uniprot.org), the function of which is largely undescribed to date.

It is not yet clear whether expression of the MS7 transcripts plays a direct or indirect role in regulating the metastatic potential of EEC, but we were able to show that higher levels of MS7 were directly correlated with *ARID1A*, *CTNNB1*, *KRAS*, *MKI67*, and *PIK3CA* and inversely correlated with *ESR1*. *CTNNB1* encodes β-catenin, *MKI67* encodes KI-67, and *ESR1* encodes ER. Functional pathway analyses provided evidence of enrichment in DNA replication and repair, inflammatory signaling, regulation of steroid hormone signaling, elevation of cell proliferation, and impairments in *TP53* signaling in patients with the highest MS7 risk scores.

## 5. Conclusions

In conclusion, our study identified and validated the MS7 score as a promising transcript-based classifier of metastasis and poor prognosis in EEC patients. Future investigations will focus on the development of robust molecular diagnostic models and evaluation of the biological and potential therapeutic relevance of the MS7 signature in independent EEC patients. The exploratory finding that the MS7 metastasis score was associated with worse PFS and OS in a cohort of EEC patients from TCGA with stage I–IV disease requires validation. The relationship between MS7 and PFS held up to adjustments for stage, grade 3 disease, and MI, indicating that MS7 was independent of these clinical factors. The development of more accurate methods for prediction of nodal and distant disease will lead to improved patterns of referral to gynecologic oncologists, better guide staging, and reduce both overtreatment and undertreatment of disease. This study provides evidence regarding the promise of molecular features and clinical factors with independent prognostic value as promising companion decision support tools for gynecologic oncology. Ultimately, we hope that companion risk assessment tools such as this will further enhance personalized, cost-effective care for EEC patients, reducing both undertreatment and overtreatment and improving outcomes for a disease with increasing incidence and mortality, as well as persistent disparities [1].

## Figures and Tables

**Figure 1 cancers-14-04070-f001:**
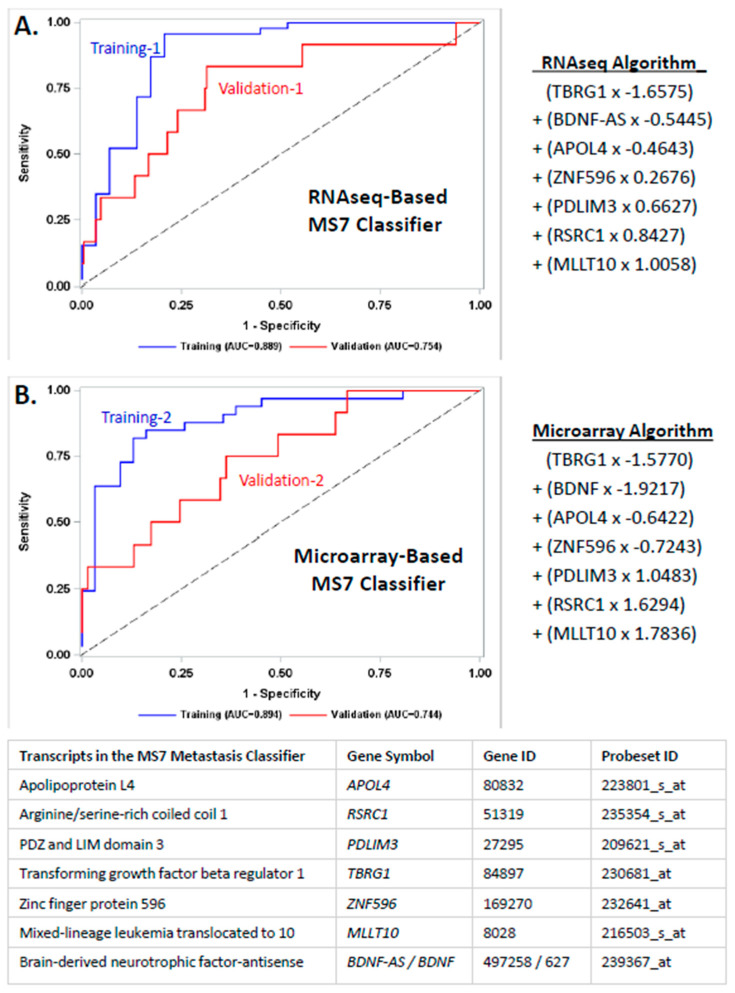
Received operator characteristic (ROC) curves for the sensitivity and 1-specificity for prediction of nodal and distant metastasis using the RNAseq-based MS7 score in the Training-1 cohort (blue line) and the Validation-1 (red line) cohort (**A**) or using the microarray-based MS7 score in the Training-2 cohort (blue line) or the Validation-2 (red line) cohort (**B**). The MS7 score integrated transcript expression data for apolipoprotein L4 (APOL4), myeloid/lymphoid or mixed-lineage leukemia translocated to 10 (MLLT10), PDZ and LIM domain 3 (PDLIM3), arginine/serine-rich coiled coil 1 (RSRC1), transforming growth factor beta regulator 1 (TBRG1), zinc finger protein 596 (ZNF596), and brain-derived neurotrophic factor antisense (BDNF-AS) using the platform-centric algorithms. The formula that was used to calculate the RNAseq algorithm and the microarray algorithm are provided to the right of the ROC curves. The Gene ID and Affymetrix probe set ID are incorporated for each transcript in the table insert below panel B.

**Figure 2 cancers-14-04070-f002:**
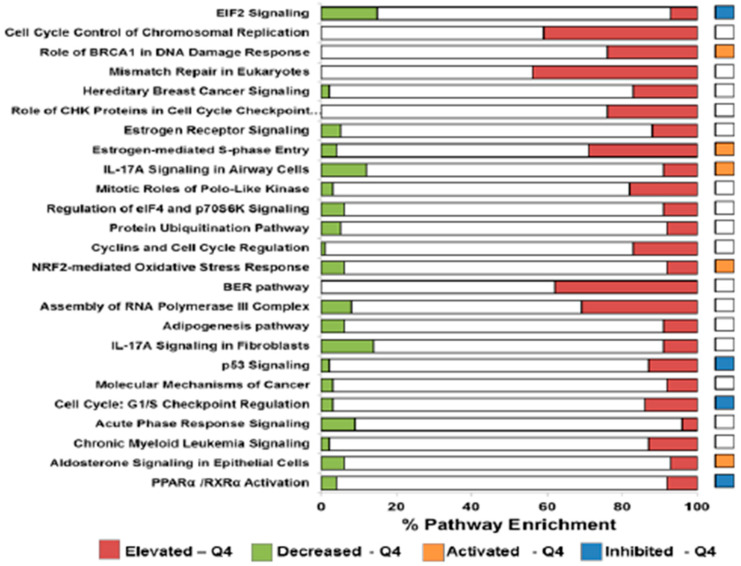
Pathway analyses of differentially expressed transcripts identified in endometrioid endometrial cancer (EEC) patients classified by high versus low metastatic risk. Data reflect the top twenty-five significant canonical pathways (–log *p*-value ≥ 2.0, Appendix A) enriched among significant differentially expressed transcripts (Appendix A) identified in RNA-seq data from endometrioid endometrial cancer patients in the Training-1 cohort classified by high (Q4, n = 19) versus low (Q1, n = 19) metastatic risk (Appendix A).

**Figure 3 cancers-14-04070-f003:**
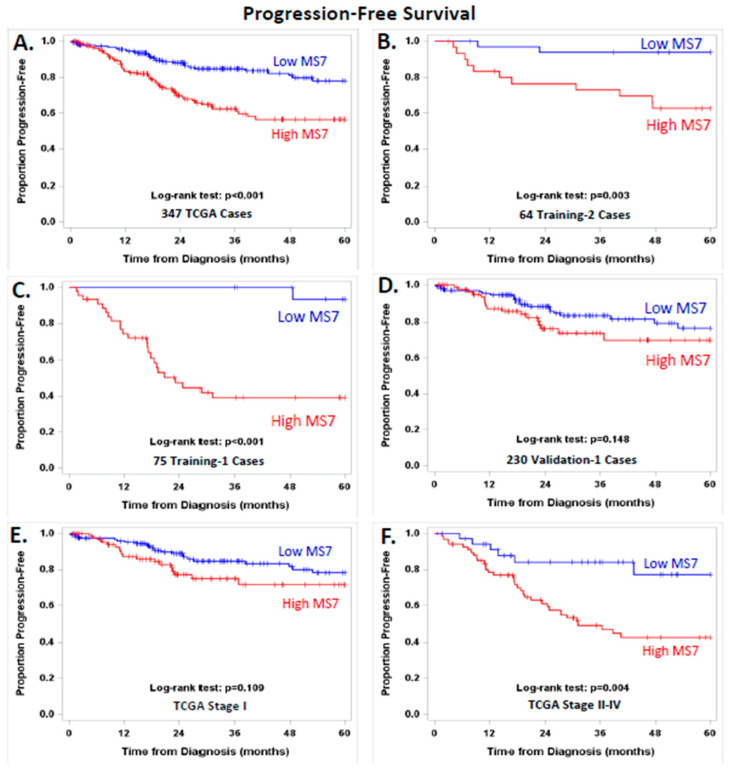
Progression-free survival plots were generated using the Kaplan–Meier method, and survival distributions were compared using log-rank tests for women with low (blue line) vs. high (red line) MS7 scores in 347 EEC patients with stage I–IV disease from the Cancer Genome Atlas Research Network (TCGA) (**A**); 64 EEC patients in the Training-2 cohort with stage I, IIIC, or IV disease (**B**); 75 EEC patients in the Training-1 cohort with stage I, IIIC, or IV disease (**C**); 230 EEC cases from the Validation-1 cohort with stage I, IIIC or IV disease (**D**); EEC patients from TCGA with stage I disease (**E**); and EEC patients from TCGA with stage II–IV disease (**F**). MS7 scores were categorized using the platform-centric cut points (−2.00356 for the RNAseq-based algorithm and −4.25324 for the microarray-based algorithm).

**Figure 4 cancers-14-04070-f004:**
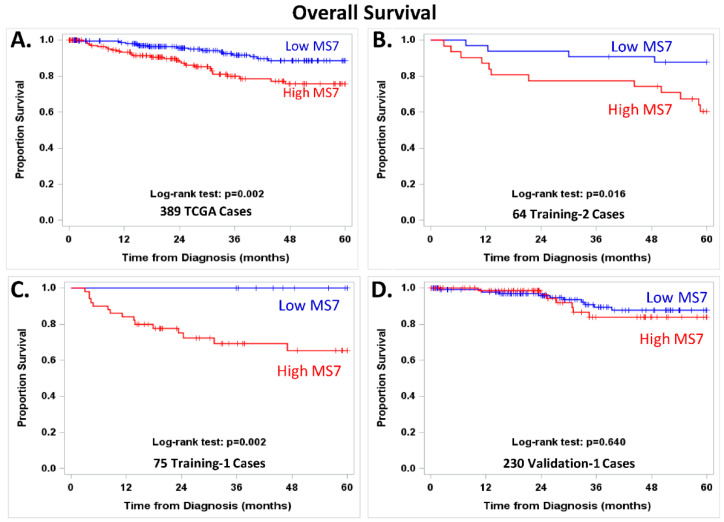
Overall survival plots were generated using the Kaplan–Meier method, and survival distributions were compared using log-rank tests for women with low (blue line) vs. high (red line) MS7 scores in 389 EEC patients with stage I–IV disease from the Cancer Genome Atlas Research Network (TCGA) (**A**); 64 EEC patients in the Training-2 cohort with stage I, IIIC, or IV disease (**B**); 75 EEC patients in the Training-1 cohort with stage I, IIIC, or IV disease (**C**); and 230 EEC cases from the Validation-1 cohort with stage I, IIIC, or IV disease (**D**). MS7 scores were categorized using the platform-centric cut points (−2.00356 for the RNAseq-based algorithm and −4.25324 for the microarray-based algorithm). The patients from the GYN-COE consortium in the Validation-1 and Valdiation-2 cohorts did not have follow-up data and were not included in the exploratory evaluation of MS7 and clinical outcome.

**Table 1 cancers-14-04070-t001:** Clinical characteristics by source for the endometrioid endometrial cancer (EEC) patients with stage I, IIIC or IV disease and RNA sequencing data or Affymetrix Plus 2.0 microarray data in the primary analysis or with stage I–IV disease and RNA sequencing data for the exploratory analyses.

	RNA Sequencing Data			Affymetrix Microarray Data	
	*Training-1*	*Validation-1*	*Exploratory*	*Training-2*	*Validation-2*
Median Age in Years	61	63	62	61.8	-
[Interquartile Range]	[55.0–67.0]	[55.0–71.0]	[55.0–70.0]	[56.5–72.8]	
<60	32 (42.7)	86 (37.9)	154 (39.9)	27 (42.2)	-
≥60	43 (57.3)	141 (62.1)	232 (60.1)	37 (57.8)	-
Unknown	*-*	*18*	*3*	*-*	*81*
Tumor Grade					
1 (G1)	13 (17.3)	65 (26.5)	93 (23.9)	12 (18.7)	26 (32.1)
2 (G2)	14 (18.7)	77 (31.4)	106 (27.3)	15 (23.4)	33 (40.7)
3 (G3)	48 (64.0)	103 (42.0)	190 (48.8)	37 (57.8)	22 (27.2)
Stage					
IA	24 (32.0)	163 (66.5)	185 (47.6)	18 (28.1)	53 (65.4)
IB	5 (6.7)	70 (28.6)	91 (23.4)	13 (20.3)	16 (19.8)
II	-	-	31 (8.0)	-	-
III/IIIA/IIIB	-	-	31 (8.0)	-	-
IIIC	33 (44.0)	10 (4.1)	37 (9.5)	22 (34.4)	9 (11.1)
IV	13 (17.3)	2 (0.8)	14 (3.6)	11 (17.2)	3 (3.7)
Myometrial Invasion					
<50%	43 (59.7)	166 (69.8)	245 (70.2)	27 (42.2)	56 (69.1)
≥>50%	29 (40.3)	72 (30.3)	104 (29.8)	37 (57.8)	25 (30.9)
Unknown	3	7	40	-	-
Metastasis Status					
Uterine-Confined Disease	29 (38.7)	233 (95.1)	276 (71.0)	31 (48.4)	69 (85.2)
Other Metastatic State	-	-	62 (15.9)		
Nodal/Distant Metastasis	46 (61.3)	12 (4.9)	51 (13.1)	33 (51.6)	12 (14.8)
Source					
TCGA	75	230	389	-	-
GOG	-	-	-	64	-
GYN-COE	-	15	-	-	81

Staging criteria for stage I cases in training was stricter than those in validation cohorts (see Methods for details). Percentage given in parentheses. The Cancer Genome Atlas Project (TCGA), Gynecologic Oncology Group (GOG), Gynecologic Cancer Center of Excellence (GYN-COE). TCGA evaluated intact frozen primary tumor samples. The GYN-COE evaluated enriched frozen tumor samples following micro or macro dissection from the GOG or GYN-COE consortium sites. There are an additional 62 cases with endometrioid endometrial cancer from TCGA with stage II, III not otherwise specified, IIIA or IIIB disease that were used for exploratory analyses. Of the 389 TCGA cases available for exploratory outcome analysis, progression-free survival was evaluated in a subset of 347 cases and overall survival was performed in the 389 patients.

**Table 2 cancers-14-04070-t002:** Predicting nodal and distant metastasis in endometrioid endometrial cancer cohorts *.

Cohort	Prediction Model ^	AUC	95% CI
Training-1	MS7 Score	**0.89**	**0.80–0.98**
	G3 (yes/no)	**0.66**	**0.55–0.77**
	MI (yes/no)	**0.69**	**0.59–0.80**
	MS7 Score + G3 (yes/no)	**0.89**	**0.80–0.98**
	MS7 Score + MI (yes/no)	**0.92**	**0.85–0.99**
	MS7 Score + MI (yes/no) + G3 (yes/no)	**0.92**	**0.85–0.99**
Validation-1	MS7	**0.75**	**0.60–0.91**
	G3 (yes/no)	**0.67**	**0.54–0.81**
	MI (yes/no)	**0.72**	**0.58–0.86**
	MS7 + G3 (yes/no)	**0.76**	**0.59–0.92**
	MS7 + MI (yes/no)	**0.81**	**0.71–0.92**
	MS7 + MI (yes/no) + G3 (yes/no)	**0.83**	**0.72–0.94**
Training-2	MS7	**0.89**	**0.81–0.97**
	G3 (yes/no)	0.56	0.44–0.68
	MI (yes/no)	**0.65**	**0.54–0.77**
	MS7 + G3 (yes/no)	**0.90**	**0.82–0.98**
	MS7 + MI (yes/no)	**0.91**	**0.84–0.98**
	MS7 + MI (yes/no) + G3 (yes/no)	**0.92**	**0.86–0.99**
Validation-2	MS7	**0.74**	**0.59–0.90**
	G3	**0.73**	**0.59–0.88**
	MI	**0.76**	**0.62–0.90**
	MS7 + G3 (yes/no)	**0.80**	**0.66–0.94**
	MS7 + MI (yes/no)	**0.86**	**0.77–0.96**
	MS7 + MI (yes/no) + G3 (yes/no)	**0.87**	**0.78–0.97**

* Models made up of one, two or three variables were evaluated based on their accuracy in predicting nodal and distant metastasis using area under the curve (AUC) and 95% confidence interval (CI) from a receiver operating characteristic curve. Bolding was used to highlight significant relationships with *p*-value < 0.05. ^ The MS7 score was calculated using the platform-centric algorithm presented in Methods and evaluated per unit increase in the score. Evaluations were also performed for grade 3 (G3) disease and/or ≥50% myometrial invasion (MI).

**Table 3 cancers-14-04070-t003:** The predictive accuracy of different companion diagnostic models for nodal and distant metastasis in endometrioid endometrial cancer patients with either stage I or stage IIIC/IV disease.

Predictive Accuracy		RNA Sequencing Data	Affymetrix Microarray Data	Merged Data
		Validation-1 [N = 245]	Validation-2 [N = 81]	Validation 1 + 2 [N = 326]
**MS7**	SN; 95% CI	0.83; 0.52–0.98	0.75; 0.43–0.95	0.79; 0.58–0.93
	SP; 95% CI	0.61; 0.55–0.68	0.64; 0.51–0.75	0.62; 0.56–0.67
	PPV; 95% CI	0.28; 0.21–0.33	0.27; 0.18–0.37	0.27; 0.22–0.32
	NPV; 95% CI	**0.95; 0.85–1.00**	**0.93; 0.87–1.00**	**0.94; 0.90–0.98**
**MS7 + Grade 3 ^†^**	SN; 95% CI	0.92; 0.62–1.00	0.83; 0.52–0.98	0.88; 0.68–0.97
	SP; 95% CI	0.42; 0.36–0.49	0.55; 0.43–0.67	0.45; 0.40–0.51
	PPV; 95% CI	0.22; 0.18–0.25	0.25; 0.18–0.32	0.22; 0.19–0.25
	NPV; 95% CI	**0.97; 0.90–1.00**	**0.95; 0.88–1.00**	**0.95; 0.90–1.00**
**MS7 +** **MI ^‡^**	SN; 95% CI	1.00; 0.72–1.00	1.00; 0.74–1.00	1.00; 0.85–1.00
	SP; 95% CI	0.48; 0.41–0.55	0.52; 0.40–0.64	0.49; 0.43–0.55
	PPV; 95% CI	0.25; 0.23–0.28	0.27; 0.23–0.33	0.26; 0.24–0.28
	NPV; 95% CI	**1.00; 1.00–1.00**	**1.00; 1.00–1.00**	**1.00; 1.00–1.00**

SN: sensitivity, SP: specificity, PPV: positive predictive value, NPV: negative predictive value, CI: confidence interval, MS7 low or high, grade 3 (G3) yes or no, myometrial invasion (MI) < 50% or ≥50%. Platform-centric thresholds (cut points) for optimal metastasis classification were defined in the corresponding training cohort using Youden index with 95% CI for sensitivity or specificity estimated using an exact method, the 95% CI for the PPV or NPV estimated using bootstrapping with 3000 repeats. PPV and NPV estimates were calculated based on an estimated prevalence of endometrioid endometrial cancer patients having vs. not having nodal or distant metastasis of 15% vs. 85%. The threshold for the RNA sequencing- vs. the microarray-based MS7 score was −2.00356 vs. −4.25324, respectively. Predictive accuracy was also evaluated in a merged cohort of validation-1 and -2 included 326 patients with 24 metastases. † The model based on low MS7 and low grade classified 137 of 140 cases as negative for nodal or distant metastasis. ‡ The model based on low MS and <50% MI successfully classified all 145 cases as negative for nodal or distant metastasis.

## Data Availability

Transcript and clinical data for this study are accessible from https://www.ncbi.nlm.nih.gov/geo/query/acc.cgi?acc=GSE120490 and from https://gdc.cancer.gov/.

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
