# Peer review of "Improving Risk Assessment for Metastatic Disease in Endometrioid Endometrial Cancer Patients Using Molecular and Clinical Features: An NRG Oncology/Gynecologic Oncology Group Study"

_cancers, 2022, doi:10.3390/cancers14174070_

Round 1

Reviewer 1 Report

This is a well executed study to identify the prognostic significance of a transcript classifier in  endometrioid endometrial carcinomas. 

Authors describe most of the limitations of their study, however they should also comment on the unproportionally low number of stage III/IV cases in the validation cohorts. 

In addition, authors should comment on the clinical utility of their classifier. Its predictive and prognostic significance seems limited in stage I cases, that includes the majority of EECs. These patients have favourable prognosis and low probability of recurrence and authors should discuss how the classifier could assist clinical decision making

Finally, authors should comment on the proper staging of all patients enrolled in the cohorts used. 

Author Response

  • This is a well-executed study to identify the prognostic significance of a transcript classifier in endometrioid endometrial carcinomas. Authors describe most of the limitations of their study, however they should also comment on the unproportionally low number of stage III/IV cases in the validation cohorts.

RESPONSE:  The most important aspect of the study was to identify a classifier that reliably identified clinical stage I without recurrence given that patients with stage III and IV disease often have clinical findings suggestive of disease spread prior to surgery.   This point has been added to the discussion. 

  • In addition, authors should comment on the clinical utility of their classifier. Its predictive and prognostic significance seems limited in stage I cases, that includes most EECs. These patients have favorable prognosis and low probability of recurrence and authors should discuss how the classifier could assist clinical decision making

RESPONSE:   As noted above, use of the MS7 classifier is quite important for clinical stage I patients.  Optimizing detection of patients with disease (sensitivity) and ensuring that patients with a negative test are without metastasis (negative predictive value).  MS7+G3 outperformed grade alone in determining risk of metastasis.  This could incrementally be valuable for clinicians in rural setting where subspecialty gynecologic oncology services may not be available.  In a recent report by Canadian investigators, they emphasized the need to molecular biomarkers to better inform providers regarding risk and to improve practice variation not only in rural cancer centers but centralized treatment centers3,4.   This point has been added to the discussion

  • Finally, authors should comment on the proper staging of all patients enrolled in the cohorts used.

RESPONSE:  Our methods section reads “The stage I controls in the Training-1 and Training-2 Cohorts were required to have undergone strict pelvic and para-aortic lymph node sampling and to be alive at last contact with no evidence of disease after three or more years of follow up. The criteria for lymph node sampling was adopted from the GOG-210 protocol and required histologic evaluation of at least four left and four right pelvic lymph nodes, and one left and one right para-aortic lymph node.  In contrast, the eligibility criteria for the stage I cases in the two Validation Cohorts were less stringent for practical reasons and accepted controls that did not satisfy the strict criteria required for the Training Cohorts.”   

PFS Kaplan Meier curves highlighted very few recurrences after three years acknowledging that the likelihood of “missed metastasis” is unlikely even though the Validation Cohorts did not have detailed data regarding staging.

Reviewer 2 Report

In this manuscript, the authors sought to identify a unique gene signature that can predict metastatic ability of endometrial cancers. The authors found a set of seven transcripts that exhibited best prediction of metastasis. The results from this work could be of importance due to the urgent need to detect and control endometriosis. The strengths of the manuscript include independent validation of their model, in-depth analysis of the correlation of the identified classifier will clinical and genetic features of EECs, and clear discussion of the limitations of the study. The suggestions below will help the readers to place the results into the context of known features of EECs:

(1)  Include a brief discussion of the molecular features of endometrial cancer in the Introduction.

(2)  It would help the readers to utilize the gene expression datasets if these are included as separate Excel files.

(3)  Include the pathway analysis shown in Figure S13 into the main figure as it will be important to emphasize the pathways that correlate with metastasis in endometrioid endometrial cancers.

(4)  Can any of the seven transcripts independently predict the metastatic potential? Can the authors comment on which of these seven transcripts is a driver for the association with metastatic potential?

(5)  Does MS7 classifier predict the time of metastasis? Can the authors include a Kaplan Meier plot for proportion of metastasis-free survival?

(6)  Move the discussion of the known role(s) of the transcripts in MS7 from Section S16 to main text.

Author Response

In this manuscript, the authors sought to identify a unique gene signature that can predict metastatic ability of endometrial cancers. The authors found a set of seven transcripts that exhibited best prediction of metastasis. The results from this work could be of importance due to the urgent need to detect and control endometriosis. The strengths of the manuscript include independent validation of their model, in-depth analysis of the correlation of the identified classifier will clinical and genetic features of EECs, and clear discussion of the limitations of the study. The suggestions below will help the readers to place the results into the context of known features of EECs:

  • Include a brief discussion of the molecular features of endometrial cancer in the Introduction.

RESPONSE:  A statement has been added to the introduction describing molecular alterations associated with EEC in TCGA.

  • It would help the readers to utilize the gene expression datasets if these are included as separate Excel files. 

RESPONSE:  the data has already been loaded up to the NCBI Gene Expression Omnibus (GEO) and will be publicly available to readers once our manuscript has been accepted for publication.   We can additionally provide an excel table to support the request but would advocate that it not be rolled into the already lengthy supplement

  • Include the pathway analysis shown in Figure S13 into the main figure as it will be important to emphasize the pathways that correlate with metastasis in endometrioid endometrial cancers.  If it’s going to be a main figure, we may need a more high-resolution figure

RESPONSE:  SF13 has been added to the main figures as suggested by the reviewers and is called out as F3.

  • Can any of the seven transcripts independently predict the metastatic potential? Can the authors comment on which of these seven transcripts is a driver for the association with metastatic potential?

RESPONSE:   As noted in supplements 1-4, we distilled down predictive biomarkers from 1,630 transcripts noted to be significantly different between stage I EEC tumors vs those with stage III or IV disease.   S4 lists the 23 transcripts with concordant coefficients from univariate logistic regression in both Training-1 and Training-2 cohorts.  Combinatorial analysis determined which subset of biomarkers served as the drivers for prediction and MS7 is what was derived.   We have reworded the text to read “Using 10-fold cross-validation in the Training Cohorts. an exhaustive combinatorial analysis of the top 23 transcripts was performed to identify the subset of biomarkers that provided the best prediction parameters.”  We have also included a table for the reviewer and editor that notes the prediction accuracy of each transcript individually as well as the composite MS7 panel for each of the Testing and Validation cohorts (see below). 

Predicting nodal and distant metastasis in endometrioid endometrial cancer cohorts

Cohort

Prediction Model ^

AUC

95% CI

 RNA Sequencing Data

Training-1

TBRG1

0.70

0.58-0.83

BDNF

0.71

0.58-0.83

APOL4

0.69

0.57-0.82

ZNF596

0.67

0.55-0.80

PDLIM3

0.66

0.52-0.79

RSRC1

0.70

0.57-0.83

MLLT10

0.70

0.57-0.83

MS7 Composite Score

0.89

0.80-0.98

Validation-1

TBRG1

0.48

0.27-0.69

BDNF

0.74

0.55-0.94

APOL4

0.82

0.71-0.93

ZNF596

0.68

0.49-0.88

PDLIM3

0.64

0.51-0.77

RSRC1

0.63

0.42-0.83

MLLT10

0.46

0.27-0.66

MS7 Composite Score

0.75

0.60-0.91

Affymetrix Microarray Data

Training-2

TBRG1

0.67

0.54-0.81

BDNF

0.65

0.51-0.78

APOL4

0.66

0.53-0.80

ZNF596

0.69

0.56-0.82

PDLIM3

0.69

0.56-0.82

RSRC1

0.65

0.51-0.78

MLLT10

0.64

0.51-0.78

MS7 Composite Score

0.89

0.81-0.97

Validation-2

TBRG1

0.72

0.56-0.88

BDNF

0.59

0.41-0.78

APOL4

0.56

0.38-0.74

ZNF596

0.62

0.43-0.82

PDLIM3

0.68

0.46-0.89

RSRC1

0.58

0.40-0.79

MLLT10

0.51

0.30-0.72

MS7 Composite Score

0.74

0.59-0.90

  • Does MS7 classifier predict the time of metastasis? Can the authors include a Kaplan Meier plot for proportion of metastasis-free survival?

RESPONSE:  Metastasis free survival is not an endpoint typically used for endometrial cancer patients whereas progression free interval is more often used for both ovarian and endometrial cancer1,2.  Given that treatment typically involves hysterectomy, recurrences would always technically be considered metastatic for stage I cancer.  All other patients in our analysis had metastatic disease at the time of diagnosis and metastasis free interval wouldn’t be an appropriate endpoint.  The M7 model was not derived to specifically identify, local, vs regional vs distant metastasis.   However, the predictive model holds for both retroperitoneal as well as distant metastatic disease. 

  • Move the discussion of the known role(s) of the transcripts in MS7 from Section S16 to main text. 

RESPONSE:  This has been relocated per the reviewer’s suggestion.

  1. Markman M: Rational study endpoints in anti-neoplastic agent regulatory approval trials in the gynecologic malignancies.  Womens Health 2016;12:396-9.
  2. Herzog TJ, Armstrong DK, Brady MF, Coleman RL, Einstein MH, Monk BJ, Mannel RS, Thigpen JT, Umpierre SA, Villella JA, Alvarez RD:   Ovarian cancer clinical trial endpoints: Society of Gynecologic Oncology white paper.  Gynecol Oncol 2014;132:8-17.
  3. Jamieson A, Huvila J, Thompson EF, Leung S, Chiu D, Lum A, McConechy M, Grondin K, Aguirre-Hernandez R, Salvador S, Kean S, Samouelian V, Gougeon F, Azordegan N, Lytwyn A, Parra-Herran C, Offman S, Gotlieb W, Irving J, Kinloch M, Helpman L, Scott SA, Vicus D, Plante M, Huntsman DG, Gilks CB,  Talhouk A, McAlpine JN:  Variation in practice in endometrial cancer and potential for improved care and equity through molecular classification.  Gynecol Oncol 2022;165:201-14.
  4. Maxwell GL, Secord AA, Powell MA: The ProMisE of uniform care for endometrial cancer patients.  Gynecol Oncol.  2022;165:199-200.
